# Neural Mechanisms of Inhibition in Scientific Reasoning: Insights from fNIRS

**DOI:** 10.3390/brainsci14060606

**Published:** 2024-06-15

**Authors:** Donglin Liu, Samrah Jamshaid, Lijuan Wang

**Affiliations:** 1School of Psychology, Northeast Normal University, Changchun 130024, China; liudl947@nenu.edu.cn (D.L.); jiam093@nenu.edu.cn (S.J.); 2School of Psychology, Hainan Normal University, Haikou 571158, China

**Keywords:** neural mechanism, inhibition, misconceptions, scientific reasoning, fNIRS

## Abstract

This study examines the impact of response and semantic inhibition on scientific reasoning using fNIRS data from 30 students (15 male, 15 female). Utilizing Go/Nogo and Stroop-like tasks within a modified speeded-reasoning task, it was found that inhibition significantly influences scientific reasoning. Specifically, slower responses and lower accuracy on incongruent statements were linked to increased activity in bilateral dorsolateral prefrontal cortex (DLPFC) and pre-supplementary motor area (pre-SMA). The research shows that both DLPFC and pre-SMA are associated with overcoming misconceptions in scientific reasoning. The findings suggest that understanding inhibitory mechanisms can enhance educational strategies to improve critical thinking and scientific literacy.

## 1. Introduction

Learning science is hard [1]. Learning science demands more than just the accumulation of new information, and scientific thinking plays a role in this [2,3]. For nearly a decade, the cognitive processes behind scientific discovery and day-to-day scientific thinking have been the subject of intense investigation and speculation [4,5,6]. Understanding the nature of scientific thinking has been an important and central issue not only for our understanding of science but also for our understanding of what it is to be human. These examinations for proper scientific procedures often resulted in the confirmation of a specific type of reasoning approach, such as induction or deduction [7]. Inductive reasoning involves starting from specific premises and forming a general conclusion [8]. People’s ability to reason logically through observations is referred to as deductive reasoning, whereas inductive reasoning generates likely (but not certain) premises from particular and limited data [9,10].

Scientific reasoning is a way of thinking that is essential to all creative endeavors [11]. It is characterized by a set of cognitive processes that enable individuals to draw conclusions based on empirical evidence. Central to these processes is the ability to inhibit irrelevant information and preconceived notions [12]. For example, when people face the problem of “A single wire can light up a light bulb” [13], the naive concept formed by daily experience (people often see a wire connected to a light bulb in their daily lives) can interfere with scientific reasoning. Individuals must invoke inhibition to suppress the interference of prior knowledge in order to make correct judgments. However, inhibition is the mental ability of blocking out or ignoring information that is not important or is distracting while concentrating on the task at hand. When it comes to scientific reasoning, inhibition is very important because it makes sure that people can look at evidence without letting prior beliefs, confirmation bias, or emotional reactions get in the way [14,15,16,17]. Recent studies have revealed the crucial role of the dorsolateral prefrontal cortex (DLPFC) in stop actions and thoughts [18].

Lately, some studies suggested that response and semantic inhibition may play a major role in scientific reasoning [19,20,21,22]. Response inhibition is the capacity to suppress a dominant behavioral response [21] and usually causes an effect in tasks that do not provide a response after the suppression of a dominant response (i.e., Go/Nogo tasks and Stop-signal tasks). The Go/Nogo paradigm is a typical task adopted to measure the skill of suppressing dominant response, such as response inhibition [16]. In the task, rapid frequent responses are made to Go stimuli, with infrequent non-responses to Nogo stimuli [20]. In the Go/Nogo task, they find that the activity of the right inferior frontal gyrus (rIFG), the DLPFC, the SMA, and other regions has been generally activated [23,24,25]. On the other hand, semantic inhibition refers to the capacity to resolve interference from acquired knowledge [21] and is correlated with tasks that require individuals to suppress a dominant response and provide a subordinate response (e.g., Stroop-like tasks). The Stroop paradigm has been confirmed as the most famous interference control paradigm [26], which is extensively adopted to explore semantic inhibition [21]. In the Stroop task, conflicting information is presented simultaneously, with a response in the less salient aspect of the stimulus while the dominant aspect is inhibited [20,27]. Additionally, in the Stroop task, the focus on activity was on regions that comprised the DLPFC, VLPFC, and anterior cingulate cortex (ACC) [28,29]. Research has suggested that when students face problems relating to misconceptions, response inhibition increases the impact on response time, while semantic inhibition mainly influences the prediction of accuracy [20].

However, some studies do not support the relationship between semantic inhibition and scientific reasoning [30,31]. Kelemen et al. (2013) demonstrated that social science and physics specialists can have good performance for undergraduate students in speed-reasoning tasks regarding teleology, whereas no difference was reported in their scores on Stroop tasks [30]. Likewise, Stricker et al. (2021) indicated that after mathematical misconceptions were examined, the scores of students on Stroop tasks did not predict their performance in speed-reasoning tasks. As discovered by the above studies, the semantic inhibition measured by the Stroop task may not be the effective component of inhibition required for scientific reasoning tasks [31].

Recently, several studies have examined the specific brain areas involved in inhibition during speeded-reasoning tasks [32,33]. Researchers typically use speeded-reasoning tasks to verify the role of inhibition during scientific reasoning. Under time pressure, compared to consistent statements (i.e., one liter of water weighs more than one liter of air), participants often suffer from strong interference from misconceptions when reasoning those inconsistent statements (i.e., one pound of iron weighs more than one pound of feathers). A study has been conducted among the professors of chemistry, and their brain activation was measured during reasoning with chemical concepts. In addition to the dorsolateral prefrontal cortex (DLPFC) and ventrolateral prefrontal cortex (VLPFC), they also found the activation in the pre-supplementary motor area (pre-SMA). The DLPFC and VLPFC regions are commonly associated with the process of semantic inhibition [34,35,36], while pre-SMA is typically correlated with response inhibition. Many researchers suggested that these regions may involve semantic and response inhibition [37,38,39,40]. But the specific contributions of response inhibition and semantic inhibition during scientific reasoning remain unclear [22].

In viewing the above literature, current study hypothesized that scientific reasoning involves response inhibition, where individuals may suppress intuitive or preconceived responses to arrive at conclusions. Additionally, semantic inhibition plays a role in preventing misleading semantic associations from interfering with logical reasoning processes. We also hypothesized that different types of incongruent statements in scientific reasoning tasks may require distinct forms of inhibition, and neuroimaging studies could reveal specific brain regions associated with response and semantic inhibition during scientific reasoning tasks, providing empirical evidence for the role of inhibition in this cognitive process. While the neural mechanisms underlying scientific reasoning have been a subject of considerable interest, the role of inhibition, particularly in the context of fNIRS studies, remains a topic ripe for exploration.

To summarize, this study aimed to examine the existing body of research on the neural mechanisms of inhibition in scientific reasoning, with a particular focus on studies employing fNIRS. We investigated the cognitive processes involved in scientific reasoning and the neural substrates associated with inhibition. Additionally, we explored the impact of inhibitory control on scientific reasoning by examining how response and semantic inhibition affect the ability to navigate scientific tasks and misconceptions.

## 2. Materials and Methods

### 2.1. Participants

A total of 40 undergraduates voluntarily participated in this study. All the participants were recruited from science majors (e.g., mathematics, physics, chemistry, and biology). Participants were tested one week before the experiment started. Their knowledge of scientific concepts was tested by using the questions on 12 relevant disciplines. However, 10 disciplines (i.e., astronomy, fraction, physiology, germs, evolution, genetics, matter, thermodynamics, mechanic, and waves) were adopted from Shtulman and Valcarcel’s (2012) study [41], and additionally, two more disciplines (i.e., electricity and chemistry) were added in this study. Questions were based on true–false options, and the total number of questions was 60 with 120 points. A total of 30 students (15 males and 15 females; *M_age_* = 20.9, *SD_age_* = 2.267) were able to pass the pre-test (*M_score_* = 99.375, *SD_score_* = 8.957). Before starting the experiment, the participants were randomly assigned into two sets of tasks to ensure a close ratio of their knowledge level (Group 1: *M_score_* = 99.25, *SD_score_* = 9.121; Group 2: *M_score_* = 99.5, *SD_score_* = 8.789), gender and age (Group 1: 7 men and 8 women, *M_age_* = 20.85, *SD_age_* = 1.768; Group 2: 8 men and 7 women, *M_age_* = 20.95; *SD_age_* = 2.673). Through G-power 3.1.9.7 software, we calculated that a sample of 15 people per group can detect a statistical test power (1 − β) of 0.899 at a significance level (α) of 0.05 or less and a moderate main effect (f = 0.25). Only those participants were selected who were right-handed, had no neurological conditions, and had normal or corrected vision in the absence of color blindness [42]. All participants received monetary compensation (RMB 40 yuan) after completing the experiment. Ethical guidelines were strictly followed for this study. Written informed consent was obtained from the participants and gained approval from the Ethics Committee of the authors’ university (No. 2022019).

### 2.2. Material

The current study was aimed to explore the contribution of response inhibition and semantic inhibition in scientific reasoning processes through a modified speeded-reasoning task. fNIRS and a modified version of the speeded-reasoning tasks were employed to evaluate the response inhibition and semantic inhibition involved in scientific reasoning processes. The fNIR is a non-invasive technique for studying brain functional activity by measuring changes in the concentration of oxyhemoglobin and deoxyhemoglobin [43], which can ensure high spatial accuracy and high temporal accuracy simultaneously [44]. It is well suited to investigate cortical activity, including the inhibitory-associated regions [45]. Thus, fNIRS is feasible to explore the mechanism of inhibition during scientific reasoning. The variations in hemoglobin concentration were examined using a multi-channel fNIRS device (LabNIRS, Kyoto Shimadzu Corporation, Kyoto, Japan) at three wavelengths of near-infrared light (i.e., 780, 805, and 830 nm), with a sampling rate of 20 Hz. Stimulations were presented using a Dell Flat Panel Monitor, model S3220DG, with a frequency of 60 Hz, set up specifically for this purpose. Task behavior measurements such as response time and accuracy to Go stimuli were captured using E-prime 3.0 software. All statistical analyses, including those for beta values across the regions of interest, were performed using SPSS 22.0, with the False Discovery Rate (FDR) correction method applied.

### 2.3. The Modified Version of Speeded-Reasoning Task

In this study, the speeded-reasoning task was optimized by introducing Go/Nogo and Stroop-like stimuli. After reading statements, participants were presented with Go/Nogo or Stroop-like stimuli. When presented with Go stimuli or semantically congruent Stroop-like stimuli, participants need to make judgments about previous statements. This cognitive process involved more reasoning about statements. When presented with Nogo stimuli or semantically incongruent Stroop-like stimuli, participants perform a STOP response, in which the cognitive process involves a suppressed response to the Go/Nogo or Stroop-like task. Therefore, by probing the inhibitory-associated regions, it may be possible to directly compare the inhibitory processes between scientific reasoning and STOP response.

Thus, in Go/Nogo group, the judgment of the statements was the rapid frequently and dominant response in the Go/Nogo task. When participants were presented with the Nogo stimuli, response inhibition needed to be mobilized to suppress the intuitive response of making judgments. And in the Stroop-like group, it was utilized with semantically congruent stimuli as the dominant aspect, in which participants were requested to judge the statements rapidly and frequently, while with semantically incongruent stimuli as the less salient aspect, participants were asked to mobilize semantic inhibition to suppress the semantically incongruent information and making a STOP response.

### 2.4. Task-Activated Brain Regions Associated with Inhibition

Previous studies have revealed that there is regularity of activation of DLPFC, IFG, and ACC [46], in which the DLPFC is thought to implement cognitive control, and the ACC is linked with error detection [16]. Moreover, the rIFG has a connection with the pre-SMA, and is involved in inhibition [46].

It was anticipated that under different task stimuli, the abovementioned regions may exhibit different activity patterns. By detecting the activity of the brain regions between Go and Nogo stimuli, it is likely to intuitively compare the brain activity between scientific reasoning of Go stimuli and stop response under Nogo stimuli. Due to the limitation of fNIRS in evaluating the surface of the lateral cortex, the cortical activation of ACC cannot be monitored [42]. Thus, to explore brain regions regarding both types of inhibition, PFC and part of the motor cortex were selected as regions of interest (ROIs), and DLPFC and pre-SMA were analyzed with fNIRS probes that cover the region.

### 2.5. Types of Incongruent Statements

Additionally, existing research has generally classified incongruent statements as incongruent scientific statements (scientifically true and naïvely false) and incongruent non-scientific statements (scientifically false and naïvely true) [32,35,47]. However, the information presented by incongruent scientific and non-scientific statements may be different, with the former intuitively containing more scientific information and the latter intuitively containing more non-scientific information. Accordingly, it was speculated that it may have different reasoning processes, which may be one of the reasons why existing research failed to clarify the inhibition. Based on this, to further evaluate the inhibitory mechanisms involved in different scientific reasoning processes, following the classic speeded-reasoning task, we divide statements into three types: congruent statements (scientifically and naïvely true or scientifically and naïvely false), incongruent scientific statements, and incongruent non-scientific statements. By comparing the task performance of participants between reasoning incongruent scientific and non-scientific statements, it is possible to intuitively compare the scientific reasoning processes of different statements.

Misconceptions were shown to occur frequently in these scientific domains and impeded students’ learning [22,32,48,49,50]. Following the material design of Shtulman and Valcarcel (2012) [41], the respective concept contained four types of statements as follows (see Table 1): ① scientifically and naïvely true (TT); ② scientifically and naïvely false (FF); ③ scientifically true and naïvely false (TF); ④ scientifically false and naïvely true (FT). There was a total of 12 conceptual domains, with five misconceptions in the respective domain and four types of statements in the respective concept, totaling 240 statements. The congruent statements of TT and FF served as the control condition, and the incongruent statements of TF and FT served as two detection conditions.

### 2.6. Task

This task refers to speeded-reasoning tasks [41]. Concepts from 12 domains are randomly presented, with a guide to remind participants of the domain of the group’s problem before the presence of the respective domain concept. First, at the beginning of the experiment, a 1000 ms fixation “+” appeared. Next, statements appeared, and participants have to read them as soon as possible and immediately press the “space” after reading. Subsequently, Go/Nogo stimuli (Go: 
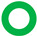
; Nogo: 
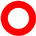
), or Stroop-like stimuli (Go: RED, GREEN; Nogo: RED, GREEN, were presented in Chinese as Go: 红, 绿; Nogo: 红, 绿) appeared. When the Go stimuli (
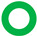
 or 红, 绿) appeared, participants immediately assessed the scientific validity of the statements (true to press the “J” key, false to press the “F” key). When the Nogo stimuli (
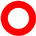
 or 红, 绿) appeared, participants were not required to respond and wait for 2000 ms before jumping on their own. Afterward, a blank screen lasting for 5000 ms appeared, marking the end of a trial (Figure 1). An exercise experiment was performed before the experiment, which comprised eight trials, and the content of the exercise experiment was not covered in the formal experiment.

### 2.7. Procedure

Before recording fNIRS, a brief explanation was provided to the participants. To avoid the fatigue of participants, they were told to take a customized break between the respective scientific domains so that they can rest until they feel rested. Participants were informed that the experimental tasks are difficult and require them to read the statements as quickly as possible and make judgments as accurately as possible. Subsequently, an exercise experiment consisting of 8 trials was conducted, and participants repeated the exercises until they mastered the task requirements. To ensure the quality of fNIRS data, participants were explicitly instructed not to move as much as possible during exercise tasks and image acquisition.

#### 2.7.1. Task Behavior Measurements

During the task, E-prime 3.0 was adopted to record the response time and accuracy of Go stimuli. These results reflect the RT and accuracy of the participant’s reaction to the statement.

#### 2.7.2. Functional Near-Infrared Spectroscopy

With the limited depth penetration of lights in fNIRS, the functional arguments of this study were limited to the prefrontal cortex and part of the motor cortex. A total of 49 channels were set up to cover the relevant regions. As depicted in Figure 2, the arrangement of 5 × 6 layout was placed in the prefrontal cortex and supplementary motor cortex, with the distance between adjacent transmitters and detectors reaching 3 cm. A 3D digitizer (FASTRAK in Boehames, VT, USA) was adopted to obtain the coordinates of all channels and rendered them to the standard brain of the Montreal Institute of Neurology using the NIRS-SPM software (SPM 8) [51] The Brodmann areas (BA) with the highest probability were found for each channel using the Talairach Daemon [52]. The optode set-up encompasses several brain regions, approximately including BA 4, BA 6, BA 8, BA 9, BA 10, and BA 46.

Due to potential variations in head size and structure, it is possible that optode localization may change. In order to enhance the accuracy of area specification, the channels, which serve as regions of interest (ROIs), were grouped into eight distinct regions [42,53,54]. The allocation of channels to regions was established based on their respective Brodmann areas (Table 2). According to previous studies, beta values were computed as an average across channels for the specific location [53,55]. Consequently, one value was ascertained for every participant within their corresponding region, provided that there were valid data available for at least one channel.

fNIRS data preprocessing. After reducing the sampling rate to 10 Hz, we used the toolbox of SPM 8 for preprocessing [56]. NIRS_ SPM was used to analyze fNIRS data, with the specific analysis process as follows:The original data were pre-processed with hemodynamic response function (HRF) [51] and wavelet minimum description length (wavelet MDL) to remove noise (such as action, heartbeat, and machine noise);The parameter estimation is performed using the general linear model (GLM) to obtain beta values, with a positive beta value indicating activation and a negative beta value indicating deactivation [57];Statistical analysis was conducted on the beta values of the respective region using SPSS [45,53], and FDR (false discovery rate) was employed for correction.

In accordance with previous studies, noisy channels were identified with a combination of objective criteria [53]. (For example, the coefficient of variance, a “flat” line detection, and visual examination). Noisy channels (about 6.8%) have been excluded from all deep analyses.

### 2.8. Statistical Analysis

First, for all the analyses, 10 participants were excluded due to scoring below 80 (full marks 120) in the pre-test. The final analyses were based on data from 30 participants (mean age: 20.9 ± 2.3 years, 15 men and 15 women). Second, we excluded trials in which the participants responded incorrectly. For measuring RTs, we included only data from trials in which participants gave correct responses to the Go stimuli. Finally, for each participant, we obtained the averaged RTs and correct rates separately for the Go stimuli items in the Go/Nogo and Stroop groups.

Using SPSS 22.0, we performed 2 (task type: Go/Nogo, Stroop-like) × 3 (statement type: TTFF, TF, FT) repeated-measures analysis of variance. The dependent variable was the response times (RTs) and accuracy of the participants to Go stimuli. We considered *p* < 0.05 to be statistically significant.

## 3. Results

### 3.1. Behavioral Results

#### 3.1.1. Response Time Results

We only analyzed the response time results for correct answers. Figure 3 visually depicts the difference in response time across statement types for both task groups. As indicated by the variance analysis of repeated measurement for Go stimuli response time, the main effect of task type was significant, with *F*(1, 28) = 37.168, *p* < 0.001, η^2^ = 0.570; the main effect of statement type was significant, with *F*(1, 28) = 16.097, *p* < 0.001, η^2^ = 0.365; the interaction between task and statement type was not significant, with *F* (1, 28) = 0.412, *p* = 0.664, η^2^ = 0.015.

As revealed by the further *t*-test results of the double samples, the reaction speed of participants under the Go/Nogo task conditions was significantly faster than that under the Stroop-like task conditions. Paired-sample *t*-tests were conducted on statement type under two task conditions, respectively (see Table 3). As indicated by the results, the Go/Nogo group displayed significant differences between FT and control conditions; and significant differences between TF and FT conditions. Furthermore, the Stroop-like group displayed significant differences between FT and control conditions, and between TF and FT conditions.

To determine the interference strength of task types on participants’ conceptual judgment of RT, the interference strength of different tasks on RT of statement types was determined by following the equation of interference strength indicator of Stricker et al. (2021) [31].
Interference_TF_ = RT_TF trials_ − RT_TTFF trials_(1)
Interference_FT_ = RT_FT trials_ − RT_TTFF trials_(2)

RT_TF trials_, RT_FT trials_, and RT_TTFF trials_ respectively represent the mean response time of each participant on TF, FT, and TTFF statements. Interference RT was calculated based on correct responses only.

As indicated by the independent-sample *t*-test for the interference strength of RT, there was a non-significant difference between TF and FT for different groups, with *t_TF_* (28) = −0.401, *p* = 0.691, *d* = 0.152; *t_FT_* (28) = 0.594, *p* = 0.557, and *d* = 0.225. The result showed that, after considering the control condition, different task types respond to response time of TF and FT conditions with the same intensity of interference.

#### 3.1.2. Accuracy Results

The accuracy of statement judgment under Go stimuli conditions. Figure 4 visually depicts the difference in accuracy across statement types for both task groups. The variance analysis of repeated measurement in terms of the accuracy of statement judgment under Go stimuli conditions shows that the main effect of task type was not significant, with *F*(1, 28) = 0.331, *p* = 0.570, η^2^ = 0.012; the main effect of statement type was significant, with *F*(1, 28) = 189.191, *p* < 0.001, η^2^ = 0.808; the interaction between task type and statement type was not significant, with *F*(1, 28) = 1.374, *p* = 0.251, η^2^ = 0.051.

The results of independent-sample *t*-test on the accuracy of statement judgment under Go stimuli conditions suggested that a non-significant difference was reported in accuracy whether in the control group (TTFF) or two incongruent conditions (TF, FT). Further paired-sample *t*-tests suggested that in the Go/Nogo group, the accuracy of participants under TF and FT conditions was significantly lower than that of the control group (TTFF), and the accuracy of participants between TF and FT conditions was significant; and in the Stroop-like group, the accuracy of participants under TF and FT conditions was also significantly lower than that of the control group, and the accuracy of participants between TF and FT conditions was also significantly (see Table 4).

### 3.2. fNIRS Results

Go/Nogo group activation analysis to determine the brain activation in the task, the beta values in the respective region of the Go stimuli, and Nogo stimuli under different statement conditions were selected as the dependent variable for a single-sample *t*-test, with a test value of 0. As indicated by the results, the brain regions related to inhibition, including the pre-SMA and the bilateral DLPFC, were significantly activated under all conditions (Figure 5).

Comparison of Different Conditions. To further test our hypothesis, we conducted paired-sample *t*-tests of significantly activated regions of interest (ROIs) to explore the response of cortical regions to statements reasoning and Nogo stimuli conditions. Results of the contrasts between the ROIs are presented in Table 5.

As indicated by the ROI comparison results, under the Nogo stimuli and FT conditions, the HbO concentration of pre-SMA was significantly increased, while when participants made judgments on the TF condition, the HbO concentration of rDLPFC significantly increased. The activation of pre-SMA increase in the Nogo stimuli condition seems to reveal the role of response inhibition in stop response during the task. Given the similarity of FT and Nogo stimuli conditions, it seems that both activate similar nervous systems.

Stroop-like group activation analysis of the identical treatment method served as the Go/Nogo group. As indicated by the results, the brain regions related to inhibition, including the pre-SMA and the bilateral DLPFC, were significantly activated under all conditions (Figure 6).

Comparison of Different Conditions. Likewise, to explore the response of cortical regions to statements reasoning and Nogo stimuli conditions, we conducted paired-sample *t*-tests on each significantly activated ROI, and the results are shown in Table 6.

As indicated by the ROI comparison results, the HbO concentration of bilateral DLPFC significantly increased under the Nogo stimuli and TF conditions, and the HbO concentration of pre-SMA significantly increased under the Nogo stimuli and TF conditions. The activation of bilateral DLPFC increase in the Nogo stimuli condition seems to reveal the role of semantic inhibition in stop response during the task. The similarity of TF and Nogo stimuli suggests that they engage similar neural systems.

Overall, our results suggested that the brain regions related to inhibition are widely activated during scientific reasoning. Compared to the control condition, the DLPFC and pre-SMA were more involved in reasoning incongruent scientific and non-scientific statements, respectively.

## 4. Discussion

In this study, behavioral methods were integrated with fNIR technology to compare the brain activity of science subject’s students across Go/Nogo and Stroop-like inhibitory tasks, and the potential inhibitory mechanisms of different scientific reasoning processes were explored. The current study hypothesized that scientific reasoning involves response inhibition, where individuals may suppress intuitive or preconceived responses to arrive at accurate conclusions. The findings of this study were significant and have supported the hypothesis. Previous studies support the study hypothesis as consist with existing research [41,58,59], lower accuracy in reasoning incongruent statements, and longer response times in reasoning incongruent non-scientific statements (FT) compared with congruent statements. Although a significant difference was not identified in response time between incongruent scientific statements (TF) and congruent statements, both groups of participants had longer response times under TF conditions (in the Go/Nogo group, 27.27 ms, (628.209 vs. 600.939 ms); in the Stroop-like group, 40.543 ms, (884.975 vs. 844.432 ms)). Since the participants in this study were all experienced experts, the decrease in accuracy was likely to indicate the persistence of misconceptions. The extension of response time under incongruent conditions may indicate additional cognitive conflicts arising from competition between misconceptions and scientific concepts, i.e., the need to suppress the interference of misconceptions when participants scientifically provide an appropriate answer to incongruent statements [22].

The fNIRS results also support the hypothesis that “participants extensively activate the brain areas associated with inhibition during scientific reasoning“, which may comprise bilateral DLPFC and pre-SMA. Although some studies have also revealed the role of the right inferior frontal gyrus (rIFG) in inhibitory control [46], recent studies have shown that the rIFG may not be related to the implementation of inhibitory control [60,61]. Our research findings also support this, as the effects of DLPFC and pre-SMA were detected during the task, rather than in the rIFG, which probably means that participants required engage inhibition when confronted with misconceptions and scientific concepts [62,63]. Although the participants activated inhibition-associated brain regions under all conditions, in comparison with congruent statements, they recruited more pre-SMA and DLPFC in reasoning incongruent statements of TF and FT, respectively. The above-described result is consistent with the fMRI study conducted by Potvin et al. (2020) [32], which reported an increase in VLPFC, DLPFC, and pre-SMA activity during chemistry professors’ reasoning incongruent statements. In brief, the results of this study support the cognitive inhibition hypothesis proposed in the present research [35,64,65].

This study also found that different types of conflicts regarding scientific problems may follow the different inhibitory mechanisms. The behavioral results indicated that participants experienced a greater impact on both response time and accuracy with FT statements. Comparing the activity regions, it is shown that rDLPFC activity increased significantly when participants were reasoning TF statements; and pre-SMA activity increased significantly when reasoning FT statements. Notably, rDLPFC typically involves top-down active control in inhibitory tasks [66] to help individuals overcome interference and choose logical responses regarding the task [36]. Thus, when reasoning TF statements, the activation of rDLPFC can be conducive to suppressing the intuitive interference of misconceptions and making judgments faster and more accurately. Pre-SMA typically involves response inhibition [23,67] and is correlated with stopping inappropriate responses/executing appropriate responses in inhibitory tasks [25]. Due to the intuitive presentation of misconceptions in the FT statements in the task, the participant may prioritize the seemingly reasonable misconception information when judging the FT statements [4], which requires stronger involvement of pre-SMA to halt the intuitive erroneous response. It should be noted that while we chose a Stroop-like task to measure semantic inhibition, it requires a certain degree of inhibition of motor responses [68], which may be one of the reasons why our task detected an increase in pre-SMA activity among Stroop group participants when responding under Nogo stimuli. Additionally, the increase in the activities of the lDLPFC only occurred in the Stroop-like task group. The possible reason for this result is that more semantic information was involved in this group of tasks, and additional lDLPFC participation is required to help the participant save conceptual information in working memory [69].

Further results showed that in the Stroop-like group, the participant showed an increase in the HbO concentration in bilateral DLPFC compared with the control condition when the Nogo stimuli and the TF condition appeared. As motioned above, DLPFC are widely activated in Stroop tasks [42,45,69], provide a top-down control, and participate in the operation of working memory information [70,71]. Subsequently, the result may suggest that the participant may have activated the similar nervous system of semantic inhibition during TF statements’ reasoning. However, in the Go/Nogo group, the Nogo stimuli and the FT condition led to an increase in the HbO concentration in pre-SMA compared with the control conditions (TTFF). Findings are consistent with the previous finding; Go/Nogo tasks are often associated with response inhibition, and pre-SMA plays a crucial role in it [23,25,67]. The above result probably suggests that the participants may have activated a similar nervous system of response inhibition during FT statements’ reasoning.

In brief, the results of this study may support the view that response inhibition and semantic inhibition play a role in addressing conflict problems [22]. Although our study cannot determine the roles of semantic inhibition and response inhibition during reasoning, it provides insights into how different types of inhibition may play different roles in scientific reasoning processes. We speculate that incongruent scientific and non-scientific statements follow different reasoning processes while involving different inhibitory components. In general, incongruent non-scientific statements intuitively present misconceptions to the participants, such that their intuitive errors can be more easily evoked. Under sophisticated task conditions, intuitive erroneous reactions of participants are more likely to be triggered, and stronger response inhibition is required to stop erroneous reactions. Compared with non-scientific statements, incongruent scientific statements involve more semantic inhibition. On that basis, the conflicts facing the participants primarily originate from the interference of prior knowledge (i.e., misconceptions) in memory [21].

This study suggests that semantic inhibition and response inhibition may play roles in different types of conflict statements. However, the findings do not indicate that participants rely solely on a specific type of inhibition when judging incongruent scientific or non-scientific statements. In fact, we suggest that semantic inhibition and response inhibition may both contribute significantly to suppressing misconceptions and selecting the correct ones in the process of resolving conflict problems. Since the significant activation of inhibition regions related to both bilateral DLPFC and pre-SMA in the reasoning process, it may be revealed the synergistic effect of different inhibitory components in addressing conflict problems. Response inhibition may contribute more to incongruent non-scientific statement reasoning, while semantic inhibition may contribute more to incongruent scientific statement reasoning. Future research can investigate the specific role of different components of inhibition during incongruent scientific and non-scientific statements reasoning.

Scientific reasoning is crucial for concept learning in education, but students often bear misconceptions even with high knowledge levels [63]. This study underscores the significance of inhibitory control as a fundamental skill in science education, similar to working memory. Enhancing inhibition abilities [22] is vital for both research and educational practice. Additionally, research has highlighted the impact of semantic inhibition on academic performance [72].

## 5. Limitations

This study faces several limitations that should be considered when interpreting its findings and designing future research. Firstly, the dependence on a sample of students may not generalize across different demographic groups. Additionally, the experimental setup, which focuses on speeded-reasoning tasks, might not accurately replicate the natural pacing of real-world scientific reasoning, potentially biasing results towards those proficient under time constraints. Moreover, the study’s narrow focus on specific types of inhibitory tasks (Go/Nogo and Stroop-like) may not capture all relevant aspects of cognitive inhibition involved in scientific reasoning, suggesting the need for broader investigatory approaches in future research. Although, this study incorporated Go/Nogo and Stroop-like stimuli to speculate on the relationship between scientific reasoning processes and inhibition. However, our results can only suggest that the activities of DLPFC and pre-SMA in scientific reasoning may be similar to these two stimuli, and we cannot conclusively conclude that semantic inhibition and response inhibition play a role in them. Further research can explore this inference in depth. Moreover, this study only focused on the semantic inhibitory effect of DLPFC in scientific reasoning. However, many studies have reported the role of VLPFC in semantic inhibition (e.g., [36], but our research did not focus on the role of VLPFC in scientific reasoning. Further research can take into account the role of VLPFC.

Furthermore, the study’s conditions of time pressure and sophisticated task/conceptual information highlight the roles of response inhibition and semantic inhibition in scientific reasoning. Teachers should consider presentation methods in science instruction to optimize inhibition processes. While multiple conceptual domains have been explored to understand the relationship between a single conceptual domain, different statements, and inhibition types, future research should focus on examining a single conceptual domain for more detailed insights. Additionally, the experimental materials used in this study had limited trials for certain concepts within a specific field. Therefore, it is recommended that future researchers increase the number of trials and reduce the number of conceptual fields to enhance the reliability and validity of the findings.

## 6. Conclusions

This study investigated the role of inhibition in scientific reasoning using a modified reasoning task and brain activation analysis. It concluded that inhibition, both in terms of response and semantic inhibition, plays a crucial role in overcoming misconceptions during scientific reasoning. Incongruent statements led to slower response times and reduced accuracy compared to congruent ones, with brain regions associated with inhibition, such as the dorsolateral prefrontal cortex and pre-supplementary motor area, showing significant activation. Interestingly, the study also revealed differences in reasoning processes and inhibitory components between incongruent scientific and non-scientific statements. Scientific statements appeared to involve more semantic inhibition, leading to quicker and more accurate responses, while non-scientific statements relied more on response inhibition, with a critical role played by the pre-SMA.

## Figures and Tables

**Figure 1 brainsci-14-00606-f001:**
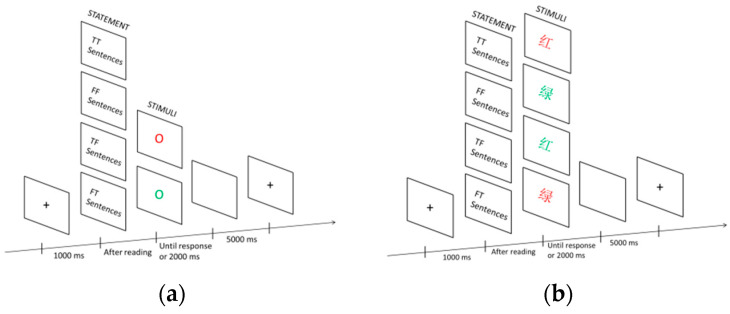
Examples of Go/Nogo group and Stroop-like group trials: (**a**) presents in the Go/Nogo group, Go stimuli and Nogo stimuli appeared after three types of statements; (**b**) presents in the Stroop-like group, two Go stimuli and two Nogo stimuli appeared after three types of statements. Participants make judgments on statements when the Go stimuli appear, whereas they do not make any response when the Nogo stimuli appear.

**Figure 2 brainsci-14-00606-f002:**
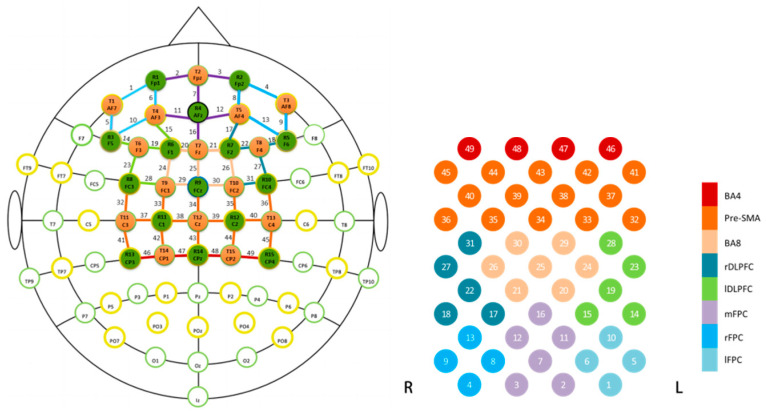
Optode localization and region specification.

**Figure 3 brainsci-14-00606-f003:**
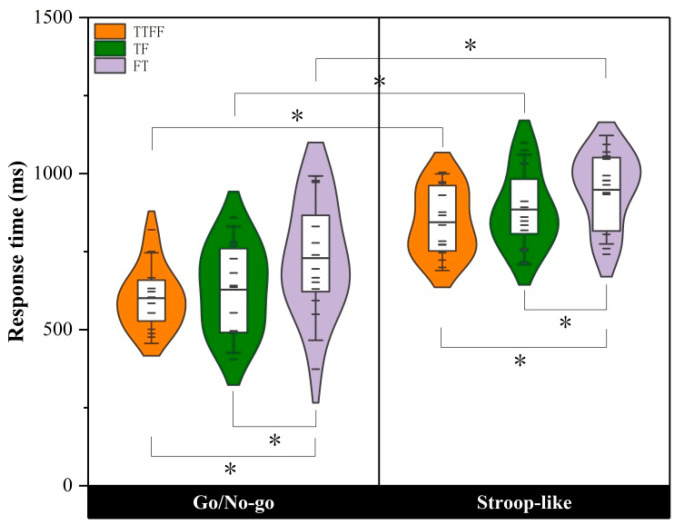
The response time of the three types of statements in the Go/Nogo group and Stroop-like group, respectively. * represent significant differences.

**Figure 4 brainsci-14-00606-f004:**
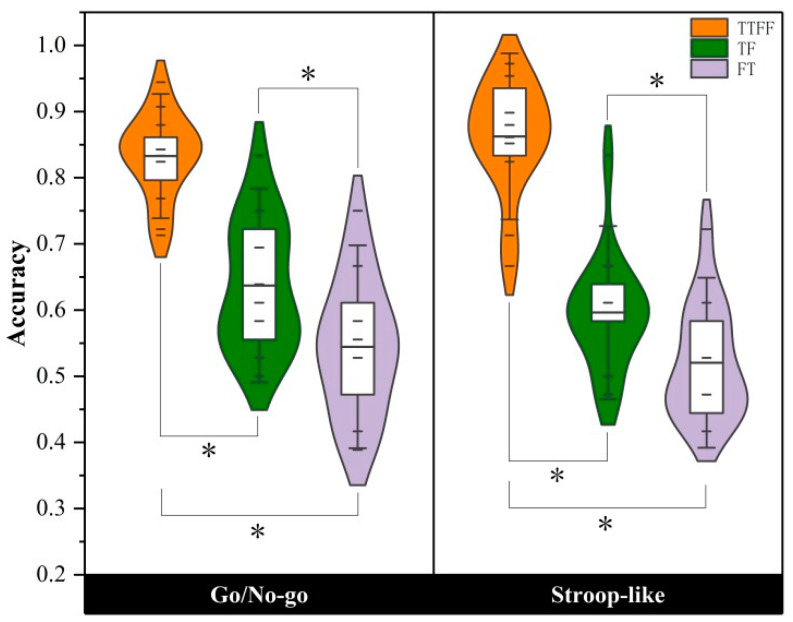
The accuracy of the three types of statements in the Go/Nogo group and Stroop-like group. * represent significant differences.

**Figure 5 brainsci-14-00606-f005:**
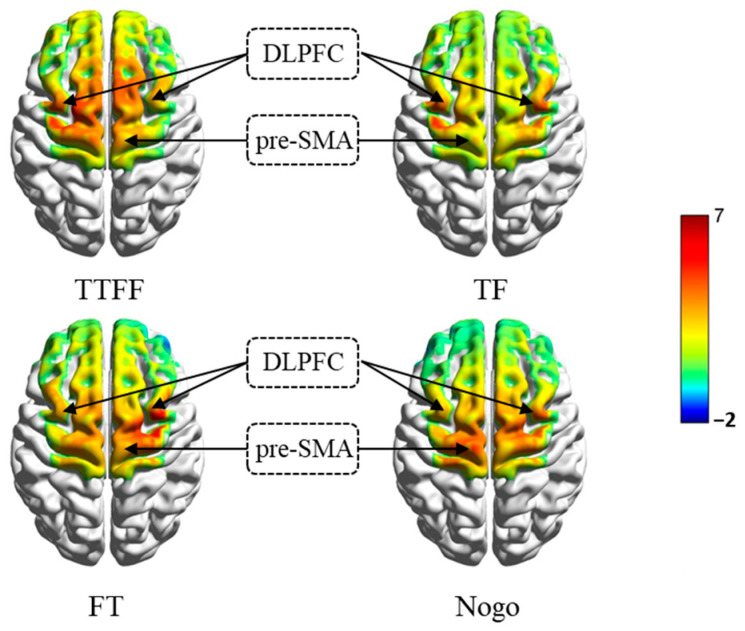
The activating region of participants in the Go/Nogo group (single-sample *t*-test).

**Figure 6 brainsci-14-00606-f006:**
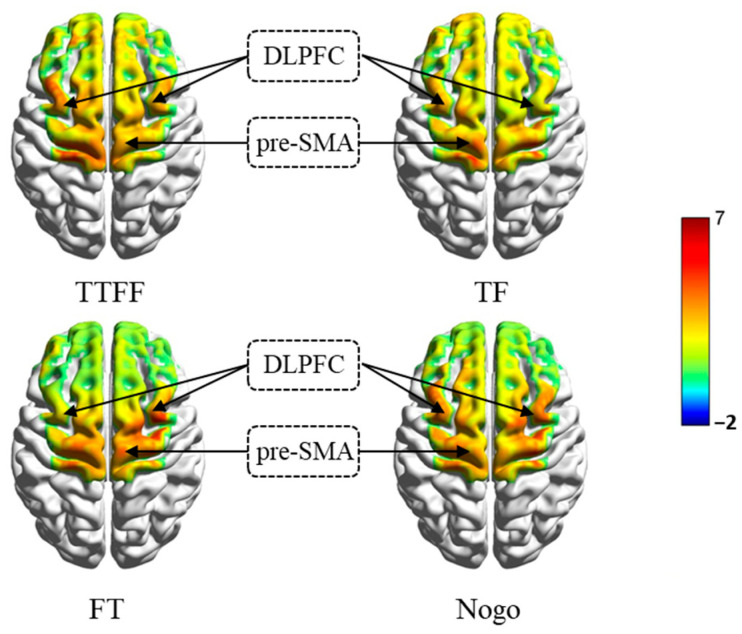
The activating region of participants in the Stroop-like group (single-sample *t*-test).

**Table 1 brainsci-14-00606-t001:** Sample items of statements. Congruent items were true or false both scientifically and naïvely; incongruent items (marked with an asterisk) were true on one theory but false on the other.

Type	Scientifically	Naïvely	Statement
TT	True	True	one liter of water weighs more than one liter of air
FF	False	False	one kilogram of iron weighs more than one ton of iron
TF	True	False	one liter of water weighs more than one liter of ice *
FT	False	True	one pound of iron weighs more than one pound of feathers *

Note: The examples provided here are in English, but the study used Chinese statements. Therefore, the translation may not be exact.

**Table 2 brainsci-14-00606-t002:** Each region of interest (ROI) corresponds to Brodmann areas and the number of channels.

Brodmann Areas	ROI	Number of Channels
BA 10	Left lateral Frontopolar Cortex (lFPC)	4 channels
	Medial Frontopolar Cortex (mFPC)	6 channels
	Right lateral Frontopolar Cortex (rFPC)	4 channels
BA 9 and BA 46	Left dorsolateral Prefrontal Cortex (lDLPFC)	5 channels
	Right dorsolateral Prefrontal Cortex (rDLPFC)	5 channels
BA 8	Includes Frontal Eye Field	7 channels
BA 6	Pre-supplementary Motor Area (pre-SMA)	14 channels
BA 4	Primary Motor Cortex	4 channels

**Table 3 brainsci-14-00606-t003:** Paired-sample *t*-tests of response time (RT) for Go stimuli of Go/Nogo and Stroop-like group.

Condition	Go/Nogo Group			Stroop-like Group		
	*t*	*p*	*d*	*t*	*p*	*d*
FT vs. TTFF	3.613	0.003 *	0.937	5.041	<0.001 *	0.980
TF vs. TTFF	1.111	0.285	0.240	1.828	0.089	0.381
TF vs. FT	−2.292	0.038 *	0.667	−2.605	0.021 *	0.564

Note. TTFF = congruent statements (control condition); TF = incongruent scientific statements; FT = incongruent non-scientific statements. * represent significant differences.

**Table 4 brainsci-14-00606-t004:** Paired-sample *t*-tests of accuracy for Go stimuli of Go/Nogo and Stroop-like group.

Condition	Go/Nogo Group			Stroop-like Group		
	*t*	*p*	*d*	*t*	*p*	*d*
FT vs. TTFF	7.944	<0.001 *	3.520	12.224	<0.001 *	4.182
TF vs. TTFF	5.717	<0.001 *	2.469	10.439	<0.001 *	3.224
TF vs. FT	3.204	0.006 *	0.959	2.929	0.011 *	0.910

Note. * represent significant differences.

**Table 5 brainsci-14-00606-t005:** Paired-sample *t*-test for Go/Nogo group participants under different conditions on each ROI.

Condition	Pre-SMA		rDLPFC		lDLPFC	
	*t*	*p*	*t*	*p*	*t*	*p*
Nogo stimuli vs. TTFF	2.882	0.012 *	1.759	0.100	0.134	0.895
TF vs. TTFF	0.132	0.897	3.024	0.009 *	0.155	0.879
FT vs. TTFFF	2.222	0.043 *	0.354	0.728	−0.201	0.844

Note. * represent significant differences.

**Table 6 brainsci-14-00606-t006:** Paired-sample *t*-test for Stroop-like group participants under different conditions on each ROI.

Condition	Pre-SMA		rDLPFC		lDLPFC	
	*t*	*p*	*t*	*p*	*t*	*p*
Nogo stimuli vs. TTFF	2.327	0.035 *	4.432	0.001 *	2.865	0.012 *
TF vs. TTFF	1.357	0.196	2.157	0.049 *	2.410	0.030 *
FT vs. TTFF	2.240	0.042 *	0.137	0.893	−0.134	0.895

Note. * represent significant differences.

## Data Availability

The raw data supporting the conclusions of this article will be made available by the authors on request.

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
