# Peer review of "Neural Mechanisms of Inhibition in Scientific Reasoning: Insights from fNIRS"

_brainsci, 2024, doi:10.3390/brainsci14060606_

Round 1

Reviewer 1 Report

Comments and Suggestions for Authors

Dear Editor,

In the manuscript entitled “Neural Mechanisms of Inhibition in Scientific Reasoning: Insights from fNIRS”, authors investigated the influence of response and semantic inhibition on scientific reasoning. The authors utilized functional Near-Infrared Spectroscopy (fNIRS) and recruited 30 students (15 male, 15 female) to participate in tasks combining Go/Nogo and Stroop-like paradigms within a modified speeded-reasoning paradigm. The results revealed an association between brain activity related to inhibition processes and behavioral parameters (e.g., accuracy) during scientific reasoning.

This is a nice study. In my opinion the paper is of potential interest for publication on Brain Sciences. However, in my view the manuscript is not ready for publication yet.

Comments:

11.      Introduction. I believe that the introduction of the manuscript should also discuss recent meta-analytical evidence (e.g., Gronchi et al., 2023) highlighting the significance of inhibitory control in reasoning. Please include a discussion of this evidence, or at the very least acknowledge that a meta-analysis supports the findings presented in this study : “When it comes to scientific reasoning, inhibition is very important because it makes sure that people can look at evidence without letting prior beliefs, confirmation bias, or emotional reactions get in the way [14-16] ”. 

Gronchi, G….. Giovannelli, F. Dual-Process Theory of Thought and Inhibitory Control: An ALE Meta-Analysis. Brain Sci. 204, 14, 101. https://doi.org/10.3390/brainsci14010101

22.     Introduction and Discussion. I believe that incorporating literature on current models of inhibitory control would enhance both the introduction and the interpretation of results in the discussion of the present study. It would be particularly valuable if the authors could draw parallels with models of response inhibition, particularly those related to pure motor paradigms, to provide further clarification of their results. Some highly relevant recent papers and metanalyses worth considering for reading and implementation are:

-Apsvalka, D., …….. Anderson, M.C., 2022. Dynamic targeting enables domain-general inhibitory control over action and thought by the prefrontal cortex. Nat. Commun. 13.

-Choo, Y., ……. Wessel, J.R., 2022. Right inferior frontal gyrus damage is associated with impaired initiation of inhibitory control, but not its implementation. Elife 11, e79667.

-Gavazzi G, ……  Viggiano MP. (2023). Subregional prefrontal cortex recruitment as a function of inhibitory demand: an fMRI metanalysis. Neurosci Biobehav Rev.

3. 3.      Material and Methods. Can the authors give more compacted details about the instrumentation employed? Like the model of the fNIRS and how they administered stimulations and  which computer? Monitor? Software or custom code in matlab/python etc. I found these information spreaded in the text… I believe that they should be reported the first time they mention the instrumentation.

44.      Materials and Methods. Please improve the quality of figure 1.

55.      Materials and methods. Authors here report a single study, It would be more convincing to report a recent meta-analysis of neural correlates: “In the Go/Nogo task, the activity of the right inferior frontal gyrus (rIFG), the dorsolateral prefrontal cortex (DLPFC), the supplementary motor area (SMA), and other regions has been generally activated [45]”

66.      Materials and methods. The same of point 5 concerning the Stroop task neural correlates.

77.      Results.For all the analyses 10 participants were excluded as they didn’t meet the criteria for
participation
.” Can you please be more specific?

88.      Results. Did the authors correct their statistical test on behavioral responses taking into account of the false discovery ratio? I have found only for neural correlates… If not please do it.

99.      Discussion. Please enrich the interpretation in Discussion according to the above point 2.

Comments on the Quality of English Language

minor editing

Author Response

Response Letter

Thank you for giving us the opportunity to submit a revised draft of the manuscript, we are highly gratified to the worthy editor and honorable reviewers for such an in-depth review/comment and for providing us a chance to improve the manuscript (brainsci-2994060, Neural Mechanisms of Inhibition in Scientific Reasoning: Insights from fNIRS) in order to provide a better piece of knowledge to the scientific community. We express our sincere gratitude for your valuable and detailed suggestions. Below, we list all of the portions of the manuscript that have been revised based on your suggestions, and we have marked these modifications in red. Line numbers refer to our inserted line numbers.

Reviewer(s)’ Comments to Author:

Reviewer: 1

Dear Editor,

In the manuscript entitled “Neural Mechanisms of Inhibition in Scientific Reasoning: Insights from fNIRS”, authors investigated the influence of response and semantic inhibition on scientific reasoning. The authors utilized functional Near-Infrared Spectroscopy (fNIRS) and recruited 30 students (15 male, 15 female) to participate in tasks combining Go/Nogo and Stroop-like paradigms within a modified speeded-reasoning paradigm. The results revealed an association between brain activity related to inhibition processes and behavioral parameters (e.g., accuracy) during scientific reasoning.

This is a nice study. In my opinion the paper is of potential interest for publication on Brain Sciences. However, in my view the manuscript is not ready for publication yet.

Comments:

1.1 – Introduction. I believe that the introduction of the manuscript should also discuss recent meta-analytical evidence (e.g., Gronchi et al., 2023) highlighting the significance of inhibitory control in reasoning. Please include a discussion of this evidence, or at the very least acknowledge that a meta-analysis supports the findings presented in this study : “When it comes to scientific reasoning, inhibition is very important because it makes sure that people can look at evidence without letting prior beliefs, confirmation bias, or emotional reactions get in the way [14-16] ”.

Gronchi, G….. Giovannelli, F. Dual-Process Theory of Thought and Inhibitory Control: An ALE Meta-Analysis. Brain Sci. 2024, 14, 101. https://doi.org/10.3390/brainsci14010101

Response: Thank you for your helpful suggestion. Based on your comment, we have cited this meta-analysis study in the manuscript to support the discussion, please refer to the updated content on Page 1, the second paragraph, line 41 – 45:

……When it comes to scientific reasoning, inhibition is very important because it makes sure that people can look at evidence without letting prior beliefs, confirmation bias, or emotional reactions get in the way [15-18].

2.2 – Introduction and Discussion. I believe that incorporating literature on current models of inhibitory control would enhance both the introduction and the interpretation of results in the discussion of the present study. It would be particularly valuable if the authors could draw parallels with models of response inhibition, particularly those related to pure motor paradigms, to provide further clarification of their results. Some highly relevant recent papers and metanalyses worth considering for reading and implementation are:

-Apsvalka, D., …….. Anderson, M.C., 2022. Dynamic targeting enables domain-general inhibitory control over action and thought by the prefrontal cortex. Nat. Commun. 13.

-Choo, Y., ……. Wessel, J.R., 2022. Right inferior frontal gyrus damage is associated with impaired initiation of inhibitory control, but not its implementation. Elife 11, e79667.

-Gavazzi G, ……  Viggiano MP. (2023). Subregional prefrontal cortex recruitment as a function of inhibitory demand: an fMRI metanalysis. Neurosci Biobehav Rev.

Response: Many thanks for your valuable suggestion. Yes, we agree with you, these literatures are very important for the explanation of the introduction and discussion. According to your suggestion, we have improved the corresponding content and cited the key literature. Please refer to the updated content on Page 1 and Page 12, the second paragraph: line 44-45, line 421-425:

……Recent studies have revealed the crucial role of the dorsolateral prefrontal cortex (DLPFC) in stop actions and thoughts.

……Although some studies have also revealed the role of the right inferior frontal gyrus (rIFG) in inhibitory control [47], recent studies have shown that the rIFG may not be related to the implementation of inhibitory control. Our research findings also support this, as the effects of DLPFC and pre SMA were detected during the task, rather than in the rIFG. ……

3.3 – Material and Methods. Can the authors give more compacted details about the instrumentation employed? Like the model of the fNIRS and how they administered stimulations and which computer? Monitor? Software or custom code in matlab/python etc. I found these information spreaded in the text… I believe that they should be reported the first time they mention the instrumentation.

Response: Thank you very much for your recommendation. We have made the amendments in the Material section. Please refer to the updated content on Page 3 to 4, second paragraph, line 143- 151, Material section:

……The variations in hemoglobin concentration were examined using a multi-channel fNIRS device (LabNIRS, Kyoto Shimadzu Corporation, Japan) at three wavelengths of near-infrared light (i.e., 780, 805, and 830 nm), with a sampling rate of 20 Hz. Stimulations were presented using a Dell Flat Panel Monitor, model S3220DG, with a frequency of 60 Hz, set up specifically for this purpose. Task behavior measurements such as response time and accuracy to Go stimuli were captured using E-prime 3.0 software. All statistical analyses, including those for beta values across the regions of interest, were performed using SPSS, with the False Discovery Rate (FDR) correction method applied.

4.4 – Materials and Methods. Please improve the quality of figure 1.

Response: Thank you for your suggestion. Due to a numbering error in the figure caption, we believe the image you are referring to might be Figure 2. We have enhanced the quality of Figure 2 accordingly. Please refer to the updated content on Page 6, line 259 Figure 2.

Figure 2. Optode localization and region specification

5.5 – Materials and methods. Authors here report a single study, It would be more convincing to report a recent meta-analysis of neural correlates: “In the Go/Nogo task, the activity of the right inferior frontal gyrus (rIFG), the dorsolateral prefrontal cortex (DLPFC), the supplementary motor area (SMA), and other regions has been generally activated [45]”

Response: Thank you very much for your valuable suggestion, we agree with you. Based on your and the Reviewer 2's comments, we have relocated this section to the introduction section and cited meta-analysis literature to support the discussion. Please refer to the updated content on page 2, the first paragraph, line 53-54:

……In the Go/Nogo task, the activity of the right inferior frontal gyrus (rIFG), the DLPFC, theSMA), and other regions has been generally activated [45]. ……

6.6 – Materials and methods. The same of point 5 concerning the Stroop task neural correlates.

Response: Thank you very much for your valuable suggestion, we agree with you. Based on your and the Reviewer 2's comments, we have relocated this section to the introduction section and cited meta-analysis literature to support the argument. Please refer to the updated content on page 2, the first paragraph, line 61-63:

……Besides, in the Stroop task, the focus on activity was on regions that comprised the DLPFC, VLPFC, and anterior cingulate cortex (ACC) [46]. ……

7.7 – Results. “For all the analyses 10 participants were excluded as they didn’t meet the criteria for participation.” Can you please be more specific?

Response: Yes, thank you for your comments. Those 10 participants did not conduct further experiments due to achieving lower scores in the scientific concept knowledge test. We revised the sentence to make it clearer. Please refer to the updated content on Page 7, line 286-287, the Statistical analysis section:

First, for all the analyses 10 participants were excluded due to scoring below 80 (full marks 120) in the pre-test. ……

8.8 – Results. Did the authors correct their statistical test on behavioral responses taking into account of the false discovery ratio? I have found only for neural correlates… If not please do it.  

Response: We appreciate your detailed suggestion. Our analysis of response time data now includes only trials with correct responses. We have revised the corresponding sections to clarify this. Please refer to the updated content on Page 7, line 289-290, the Statistical analysis section:

……For measuring RTs, we included only data from trials in which participants gave correct responses to the Go stimuli. ……

9.9 – Discussion. Please enrich the interpretation in Discussion according to the above point 2.

Response: Thanks to your comments, we added these key references to make our discussion more compelling. The revised content is included on Page 12, the second paragraph, line 421-425:

……Although some studies have also revealed the role of the right inferior frontal gyrus (rIFG) in inhibitory control [47], recent studies have shown that the rIFG may not be related to the implementation of inhibitory control. Our research findings also support this, as the effects of DLPFC and pre SMA were detected during the task, rather than in the rIFG. ……

Finally, we appreciate your constructive comments. We have gained valuable insights from them, which have significantly enhanced the quality of our manuscript.

Best regards,

Authors

Reviewer 2 Report

Comments and Suggestions for Authors Overview: This study attempts to explore the role of inhibition in scientific reasoning with fNIRS. I don't fully understand the rationale for design -- i.e., what layering Go/No-go demands, and further layering Stroop demands ("go" and "nogo" stimuli presented in red or green) on top of true/false statement judgments tells us about the processes involved in scientific reasoning. I have concerns about the way the data are analyzed and presented. No brain data are shown, even though this is a brain imaging study. Finally, I don't understand any part of how the authors arrived at their final conclusion: "Scientific statements appeared to involve more semantic inhibition, leading to quicker and more accurate responses, while non-scientific statements relied more on response inhibition, with a critical role played by the pre-SMA."   Introduction: Towards the beginning, give an example of a bias/misconception that must be suppressed in the context of scientific reasoning. Towards the end, move the text about the current study (lines 111-119) to a new paragraph and articulate the rationale & planned research more clearly. (What was unclear from the previous studies and what did you hope to address with your study? Why/how did you use both Go/No-Go-like and Stroop-like assessments of scientific knowledge?)   Methods: The electrode placements in Figure 1 look too dorsal to be BA 9/46 (superior frontal gyrus rather than the middle frontal gyrus). Double-check?   Results:

  1. Combining TT and FF (TTFF) is not appropriate for comparisons with FT and TF; this is a 2x2 factorial design. Even though the authors are not interested in the distinction between these conditions, it is helpful to see the behavioral and fNIRS results for all 4 conditions, and for these to be included as factors in the ANOVAs
  2. The bar plots and descriptions of the ANOVAs & behavioral results are quite rudimentary (beyond the fact that TT and FF are combined).
  3. Importantly, why are only 3 of the ROIs reported in Table 6?
  4. No brain data are shown - either activation at the channels or waveforms! As such it is not possible to evaluate the quality of the data.
  5. Relatedly, I would replace Tables 5 and 7 with figures.
  6. Given the lack of interaction between task and statement type on RTs, the follow-up paired t-tests conducted separately for the two types of tasks may not be appropriate.
  7. Is it possible to locate activation to pre-SMA with fNIRS?

  Discussion: VLPFC has been heavily implicated in both semantic retrieval and response inhibition (more so than DLPFC, probably) and is likely at least as relevant. Measurement of VLPFC activation is worth noting as a limitation/future direction.   Minor comments:   1. Methods: Lines 164-195: The general background about Stroop and Go/Nogo & associated brain regions should be moved to the introduction, along with anything regarding the rationale/hypotheses/approach.   2. Figure 1: "TTFF Sentences" has not been explained. Since this figure shows what participants might see on a given trial, and "TTFF" doesn't correspond to what they actually viewed on a trial, TT and FF should be displayed as separate possible stimuli.   3. Lines 95, 97: It's unclear what's meant by "the transfer effects". If this was a cognitive training study, make that clear & specify what was trained and what they sought to test transfer to.   4. Line 272-273: Clarify that the Brodmann areas are approximate   5. Lines 41-43: "Working memory, inhibitory control, and cognitive control are all executive functions that are handled by the dorsolateral prefrontal cortex (DLPFC)." Rephrase, as DLPFC does not operate on its own.

Comments on the Quality of English Language

The quality of the English language was overall quite good, although the description of the behavioral results was confusing.

Line 52: "In this literature review" implies that this is a review paper, not a study   Line 71: articulate the utility of a speeded reasoning task.   The terms "the present investigation" (line 74) and "the present study" (Line 81) are confusing because they do not refer to this paper.   Line 122: "Totally" -> "A total of"   Line 136: "had neurological normal conditions" -> "had no neurological conditions"   Line 220: "detective levels"?   Table 1: "outweighing" -> "weighs more than"   Line 406: "both activate similar nervous systems" -- rephrase

Author Response

Response Letter

Thank you for giving us the opportunity to submit a revised draft of the manuscript, we are highly gratified to the worthy editor and honorable reviewers for such an in-depth review/comment and for providing us a chance to improve the manuscript (brainsci-2994060, Neural Mechanisms of Inhibition in Scientific Reasoning: Insights from fNIRS) in order to provide a better piece of knowledge to the scientific community. We express our sincere gratitude for your valuable and detailed suggestions. Below, we list all of the portions of the manuscript that have been revised based on your suggestions, and we have marked these modifications in red. Line numbers refer to our inserted line numbers.

Reviewer(s)’ Comments to Author:

Reviewer: 2

Comments to the Author

Overview: This study attempts to explore the role of inhibition in scientific reasoning with fNIRS.

  • I don't fully understand the rationale for design -- i.e., what layering Go/No-go demands, and further layering Stroop demands ("go" and "nogo" stimuli presented in red or green) on top of true/false statement judgments tells us about the processes involved in scientific reasoning.

Response: Thank you for your comment. We believe that by adding Go/Nogo or Stroop-like stimuli after a scientific statement, it is possible to intuitively observe the changes in HbO concentration in the inhibition related regions of participants during scientific reasoning, as well as the changes in HbO concentration in the inhibition related regions of participants during Nogo stimuli (i.e., response and semantic inhibition). By directly comparing the characteristics of these two in terms of brain activity, it is possible to intuitively compare the relationship related to inhibitory region activity between scientific reasoning and the inhibition of response inhibition and semantic inhibition.

  • I have concerns about the way the data are analyzed and presented. No brain data are shown, even though this is a brain imaging study.

Response: Many thanks for your valuable suggestion. Yes, we agree with your concerns about the way the data were analyzed and presented. So we visualized tables 5 and 7 in the fNIRS results section of the previous manuscript and provided brain imaging figures. Please see the revised content on Pages 10 and 11, Figure 5 and Figure 6:

Figure 5. The activating region of participants in the Go/Nogo group (single sample t-test)

Figure 6. The activating region of participants in the Stroop-like group (single sample t-test)

  • Finally, I don't understand any part of how the authors arrived at their final conclusion: "Scientific statements appeared to involve more semantic inhibition, leading to quicker and more accurate responses, while non-scientific statements relied more on response inhibition, with a critical role played by the pre-SMA."

Response: Thank you very much for your comment. Our fNIRS results indicate that in the Go/Nogo group, compared to the control condition (TTFF), the activated regions of reasoning FT statements and the response of Nogo stimuli related to response inhibition were similar, both of which strongly activate pre-SMA; In the Stroop group, compared to the control condition (TTFF), the activated regions of reasoning TF statements and the response of Nogo stimuli related to semantic inhibition were similar, both of which activated the bilateral DLPFC more strongly. This indicates that during the reasoning process of FT statements, the activity of the inhibitory related regions (pre-SMA) may be closer to response inhibition, perhaps reflecting the role of response inhibition in this process. During the reasoning of TF statements, inhibiting the activity of relevant regions may be closer to semantic inhibition, which may reflect the role of semantic inhibition. Meanwhile, our behavioral results indicated that compared to FT statements, participants have faster response times and higher accuracy in reasoning TF statements. Combining the fNIRS and behavioral results, it seems that semantic inhibition plays a greater role during TF statement reasoning, and this type of statement has a higher inference accuracy and faster response speed.

  • Introduction: Towards the beginning, give an example of a bias/misconception that must be suppressed in the context of scientific reasoning. Towards the end, move the text about the current study (lines 111-119) to a new paragraph and articulate the rationale & planned research more clearly. (What was unclear from the previous studies and what did you hope to address with your study? Why/how did you use both Go/No-Go-like and Stroop-like assessments of scientific knowledge?)

Response: Thank you for your valuable suggestion. We have added an example of how inhibition plays a role in scientific reasoning to describe the necessity of inhibition in scientific reasoning. And clearly state the research hypothesis and aim at the end of the introduction section. Please see the revised content on Page 1, the second paragraph; and Pages 2-3, the last paragraph, line 35 -39, line 90-106:

……For example, when people face the problem of A single wire can light up a light bulb" [14], the naive concept formed by daily experience (people often see a wire connected to a light bulb in their daily lives) can interfere with scientific reasoning. Individual must invoke inhibition to suppress the interference of prior knowledge in order to make correct judgments. ……

As viewing the above literature, current study hypothesized that scientific reasoning involves response inhibition, where individuals may suppress intuitive or preconceived responses to arrive at conclusions. Additionally, semantic inhibition plays a role in preventing misleading semantic associations from interfering with logical reasoning processes. We also hypothesized that different types of incongruent statements in scientific reasoning tasks may require distinct forms of inhibition and neuroimaging studies could reveal specific brain regions associated with response and semantic inhibition during scientific reasoning tasks, providing empirical evidence for the role of inhibition in this cognitive process. While the neural mechanisms underlying scientific reasoning have been a subject of considerable interest, the role of inhibition, particularly in the context of fNIRS studies, remains a topic ripe for exploration.

To summarize, this study aimed to examine the existing body of research on the neural mechanisms of inhibition in scientific reasoning, with a particular focus on studies employing fNIRS. We investigated the cognitive processes involved in scientific reasoning and the neural substrates associated with inhibition. Additionally, we explored the impact of inhibitory control on scientific reasoning by examining how response and semantic inhibition affect the ability to navigate scientific tasks and misconceptions.

  • Methods: The electrode placements in Figure 1 look too dorsal to be BA 9/46 (superior frontal gyrus rather than the middle frontal gyrus). Double-check?

Response: Thank you very much for your valuable suggestion. We have redrawn the image in order to display our channel layout more clearly, please see the revised content on Page 6, Figure 2:

Figure 2. Optode localization and region specification

Results

1 – Combining TT and FF (TTFF) is not appropriate for comparisons with FT and TF; this is a 2x2 factorial design. Even though the authors are not interested in the distinction between these conditions, it is helpful to see the behavioral and fNIRS results for all 4 conditions, and for these to be included as factors in the ANOVAs

Response: Thank you for the suggestion. In previous studies, TT and FF statements were used as consistency conditions to compare with TF and FT statements that were inconsistent conditions. In our study, we followed the previous approach of using TT and FF statements as consistent conditions and treated them as control groups. And further consider inconsistent statements separately, dividing them into inconsistent scientific statements (TF) and inconsistent non-scientific statements (FT). By comparing the reasoning processes between control conditions and TF & FT statements, we can further explore the different processes and inhibitory effects of scientific reasoning. Thus, we don't need to consider TT and FF statements separately.

2 – The bar plots and descriptions of the ANOVAs & behavioral results are quite rudimentary (beyond the fact that TT and FF are combined).

Response: Yes, thank you for this detailed suggestion. We have updated the descriptive statistical figure of behavioral results. Please see the revised content on Pages 8 and 9, Figure 3 and Figure 4:

Figure 3. The response time of the three types of statements in the Go/Nogo group and Stroop-like group, respectively

Figure 4. The accuracy of the three types of statements in the Go/Nogo group and Stroop-like group

3 – Importantly, why are only 3 of the ROIs reported in Table 6?  

Response: Thanks to your comments. Due to our focus on inhibitory related regions, we include bilateral DLPFC and pre-SMA. Therefore, after performing fdr correction on the eight ROIs, we mainly reported the significantly activated brain regions.

4 – No brain data are shown - either activation at the channels or waveforms! As such it is not possible to evaluate the quality of the data.

Response: Thanks to your valuable comments, we have changed the previous tables to activation figures for a more intuitive observation of data quality. Please see the revised content on Pages 10 and 11, Figure 5 and Figure 6:

Figure 5. The activating region of participants in the Go/Nogo group (single sample t-test)

Figure 6. The activating region of participants in the Stroop-like group (single sample t-test)

5 – Relatedly, I would replace Tables 5 and 7 with figures.

Response: Thank you for these valuable suggestions. As we mentioned in comment 4, we have replaced the tables to figures.

6 – Given the lack of interaction between task and statement type on RTs, the follow-up paired t-tests conducted separately for the two types of tasks may not be appropriate.

Response: Thank you for your helpful suggestion. Due to the first separation of TF and FT statement types in our study. Therefore, although the interaction between task types and statement types was not significant, but the main effect of statement types may suggest that there may be differences in response time and accuracy between TF and FT statements. Therefore, we further conducted paired sample t-tests as exploratory analysis to examine the differences in response time and accuracy between TF and FT statements.

7 – Is it possible to locate activation to pre-SMA with fNIRS?

Response: Thank you. Although the central sulcus in the pre-SMA region may affect the quality of fNIRS data. But a large amount of research on fNIRS has detected pre-SMA and obtained reliable results. For example, in Hatakenaka et al.'s (2007) study, fNIRS was used to explore the brain mechanisms of motor skill learning. In this study, they used fNIRS to record the changes in HbO concentration in the sensorimotor cortex (SMC), supplementary motor area (SMA), pre-supplementary motor area (pre-SMA), and prefrontal regions during the task process.

(Hatakenaka, M., Miyai, I., Mihara, M., Sakoda, S., & Kubota, K. (2007). Frontal regions involved in learning of motor skill—A functional NIRS study. NeuroImage, 34, 109-116. https://doi.org/10.1016/j.neuroimage.2006.08.014.)

Discussion

  • VLPFC has been heavily implicated in both semantic retrieval and response inhibition (more so than DLPFC, probably) and is likely at least as relevant. Measurement of VLPFC activation is worth noting as a limitation/future direction.

Response: Thank you for your suggestion. We acknowledged the important role of VLPFC in the inhibition process. Therefore, we have added limitations on VLPFC in the limitations section, and future research should consider VLPFC. The revised content is included on Page 13, line 508-512, the Limitations section:

……Moreover, this study only focused on the semantic inhibitory effect of DLPFC in scientific reasoning. However, many studies have reported the role of VLPFC in semantic inhibition (e.g., [37]), but our research did not focus on the role of VLPFC in scientific reasoning. Further research can take into account the role of VLPFC.

Minor comments

1 – Methods: Lines 164-195: The general background about Stroop and Go/Nogo & associated brain regions should be moved to the introduction, along with anything regarding the rationale/hypotheses/approach.

Response: Thank you very much for your valuable suggestion. Based on your suggestion, this section has been rewritten and the general background about Stroop and Go/Nogo & associated brain regions have been moved to the corresponding position in the introduction. The revised content is included on Page 2, line 50-54, line 57-63, the first paragraph:

……The Go/Nogo paradigm is a typical task adopted to measure the skill of suppressing dominant response, such as response inhibition [17]. In the task, rapid frequent responses are made to Go stimuli, with infrequent non-responses to Nogo stimuli [21]. In the Go/Nogo task, the activity of the right inferior frontal gyrus (rIFG), the DLPFC, theSMA), and other regions has been generally activated [24-26]. ……

The Stroop paradigm has been confirmed as the most famous interference control paradigm [27], which is extensively adopted to explore semantic inhibition [22]. In the Stroop task, conflicting information is presented simultaneously, with a response in the less salient aspect of the stimulus while the dominant aspect is inhibited [21,28]. Besides, in the Stroop task, the focus on activity was on regions that comprised the DLPFC, VLPFC, and anterior cingulate cortex (ACC) [29,30].

2 – Figure 1: "TTFF Sentences" has not been explained. Since this figure shows what participants might see on a given trial, and "TTFF" doesn't correspond to what they actually viewed on a trial, TT and FF should be displayed as separate possible stimuli.

Response: We thank the reviewer for this detailed suggestion. According to your suggestion, we have modified Figure 1. The revised content is included on Pages 5–6, Figure 1:

(a)

(b)

Figure 1. Examples of Go/Nogo group and Stroop-like group trials: (a) presents in the Go/Nogo group, Go stimuli and Nogo stimuli appeared after three types of statements; (b) presents in the Stroop-like group, two Go stimuli and two Nogo stimuli appeared after three types of statements. Participants make judgments on statements when the Go stimuli appear, whereas they do not make any response when the Nogo stimuli appear

3 – Lines 95, 97: It's unclear what's meant by "the transfer effects". If this was a cognitive training study, make that clear & specify what was trained and what they sought to test transfer to.

Response: Thank you for your helpful suggestion. We have revised the introduction section and removed it.

4 – Line 272-273: Clarify that the Brodmann areas are approximate

Response: Thank you for the excellent suggestion. Based on your suggestion, we have revised this sentence. The revised content is included on Pages 6, the fourth paragraph, line 256-257:

……The optode set-up encompasses several brain regions, approximately including BA 4, BA 6, BA 8, BA 9, BA 10, and BA 46.

5 – Lines 41-43: "Working memory, inhibitory control, and cognitive control are all executive functions that are handled by the dorsolateral prefrontal cortex (DLPFC)." Rephrase, as DLPFC does not operate on its own.

Response: We thank the reviewer for this detailed suggestion. Combined with Q3, we have revised the introduction section and removed it.

Comments on the Quality of English Language

  • Line 52: "In this literature review" implies that this is a review paper, not a study

Response: Many thanks for your valuable suggestion. Combined with Q3 and Q5, we have revised the introduction section and removed it.

  • Line 71: articulate the utility of a speeded reasoning task.

Response: We thank the reviewer for this detailed suggestion. Based on your suggestion, we have added the explanation of the utility of a speed reasoning task. Please see Pages 2, line 75-81, the third paragraph:

Recently, several studies have examined the specific brain areas involved in inhibition during speeded reasoning task [33,34]. Researchers typically use speeded reasoning task to verify the role of inhibition during scientific reasoning. Under time pressure, compared to consistent statements (i.e., one liter of water weighs more than one liter of air), participants often suffer from strong interference from misconceptions when reasoning those inconsistent statements (i.e., one pound of iron weighs more than one pound of feathers). ……

  • The terms "the present investigation" (line 74) and "the present study" (Line 81) are confusing because they do not refer to this paper.

Response: Many thanks for your detail suggestion. We have revised this section to make the expression clearer. Please see Pages 2, line 81-84, the third paragraph:

……A study has been conducted among the professors of chemistry and their brain activation was measured during reasoning with chemical concepts. In addition to the dorsolateral prefrontal cortex (DLPFC) and ventrolateral prefrontal cortex (VLPFC), they also found the activation in the pre-supplementary motor area (pre-SMA). ……

  • Line 122: "Totally" -> "A total of"
  • Line 136: "had neurological normal conditions" -> "had no neurological conditions"
  • Line 220: "detective levels"?
  • Table 1: "outweighing" -> "weighs more than"
  • Line 406: "both activate similar nervous systems" -- rephrase

Response: Thank you for your helpful suggestion. Based on your suggestions, we have made amendments to these sections.

Finally, we appreciate your constructive comments. We have gained valuable insights from them, which have significantly enhanced the quality of our manuscript.

Best regards,

Author

Reviewer 3 Report

Comments and Suggestions for Authors

Dear authors,

The manuscript is very interesting. Some comments are necessary though. Please, see them below.

- Once you address an abbreviation, you must use the abbreviation only to describe the term. Please, double-check the entire text.

- The introduction is way too large. Sometimes the readability is impaired. Please, keep the minimal necessary to explain the rationale.

- I could notice the hypothesis, but not the aim of the study. Please, double-check and insert if it was not previously declared.

- I couldn't notice any a priori sample size calculation. Please, explain and insert in the participants' section.

- The fNIRS device must be fully described at the first time it was cited in the text. Restructure the paragraphs positioning.

- No statistical section was found. Please, explain.

- The results are mixed with the statistics, and as no previous section was organized, it was difficult to follow the section. I strongly suggest an improved organization, with declared asumption checks, tests employed to assess the significant differences, effect sizes, etc.

- The authors must be aware of all limitations of the study. I suggest you to insert a paragraph to describe and fully discuss them.

- I strongly advise you to insert the raw data into a public online repository. 

Author Response

Response Letter

Thank you for giving us the opportunity to submit a revised draft of the manuscript, we are highly gratified to the worthy editor and honorable reviewers for such an in-depth review/comment and for providing us a chance to improve the manuscript (brainsci-2994060, Neural Mechanisms of Inhibition in Scientific Reasoning: Insights from fNIRS) in order to provide a better piece of knowledge to the scientific community. We express our sincere gratitude for your valuable and detailed suggestions. Below, we list all of the portions of the manuscript that have been revised based on your suggestions, and we have marked these modifications in red. Line numbers refer to our inserted line numbers.

Reviewer(s)’ Comments to Author:

Reviewer: 1

Comments to the Author

Dear authors,

The manuscript is very interesting. Some comments are necessary though. Please, see them below.

1 – Once you address an abbreviation, you must use the abbreviation only to describe the term. Please, double-check the entire text.

Response: Thank you for your helpful suggestion. We have double-checked the entire text and replaced all the terms with abbreviations.

2 – The introduction is way too large. Sometimes the readability is impaired. Please, keep the minimal necessary to explain the rationale.

Response: Many thanks for your valuable suggestion. Yes, we agree with you. Based on your feedback, we have made modifications to the introduction section and removed unnecessary content. Please see the revised content on Pages 1—3, the Introduction section.

3 – I could notice the hypothesis, but not the aim of the study. Please, double-check and insert if it was not previously declared.

Response: Thank you very much for your suggestion. As stated in Q2, we have made modifications to the introduction section and removed unnecessary content. Please see the revised content on Pages 2—3, the last paragraph, line 90-106:

As viewing the above literature, current study hypothesized that scientific reasoning involves response inhibition, where individuals may suppress intuitive or preconceived responses to arrive at conclusions. Additionally, semantic inhibition plays a role in preventing misleading semantic associations from interfering with logical reasoning processes. We also hypothesized that different types of incongruent statements in scientific reasoning tasks may require distinct forms of inhibition and neuroimaging studies could reveal specific brain regions associated with response and semantic inhibition during scientific reasoning tasks, providing empirical evidence for the role of inhibition in this cognitive process. While the neural mechanisms underlying scientific reasoning have been a subject of considerable interest, the role of inhibition, particularly in the context of fNIRS studies, remains a topic ripe for exploration.

To summarize, this study aimed to examine the existing body of research on the neural mechanisms of inhibition in scientific reasoning, with a particular focus on studies employing fNIRS. We investigated the cognitive processes involved in scientific reasoning and the neural substrates associated with inhibition. Additionally, we explored the impact of inhibitory control on scientific reasoning by examining how response and semantic inhibition affect the ability to navigate scientific tasks and misconceptions

4 – I couldn't notice any a priori sample size calculation. Please, explain and insert in the participants' section.

Response: Thank you for your suggestion. We have added a description of the effect size in the participants section, please see the revised content on Page 3, line 123-125, the Participants section:

……Through G-power 3.1.9.7 software, we calculated that a sample of 15 people per-group can detect a statistical test power (1- β) of 0.899 at a significance level (α) of 0.05 or less and a moderate main effect (f = 0.25). ……

5 – The fNIRS device must be fully described at the first time it was cited in the text. Restructure the paragraphs positioning.

Response: Thank you very much for your valuable suggestion. We have modified this section to make it clearer. Please see the revised content on Pages 3—4, line 142-150,  the Material section:

……The variations in hemoglobin concentration were examined using a multi-channel fNIRS device (LabNIRS, Kyoto Shimadzu Corporation, Japan) at three wavelengths of near-infrared light (i.e., 780, 805, and 830 nm), with a sampling rate of 20 Hz. Stimulations were presented using a Dell Flat Panel Monitor, model S3220DG, with a frequency of 60 Hz, set up specifically for this purpose. Task behavior measurements such as response time and accuracy to Go stimuli were captured using E-prime 3.0 software. All statistical analyses, including those for beta values across the regions of interest, were performed using SPSS, with the False Discovery Rate (FDR) correction method applied.

6 – No statistical section was found. Please, explain.

Response: Thank you for the excellent suggestion. We revised the corresponding part of the manuscript and added a "Statistical analysis" section to better explain the data analysis software and methods we used. The Statistical analysis section is included on page 7, line 284-295:

Statistical analysis

First, for all the analyses 10 participants were excluded due to scoring below 80 (full marks 120) in the pre-test. The final analyses were based on data from 30 participants (mean age: 20.9 ± 2.3 years, 15 men and 15 women). Second, we excluded trials in which the participants responded incorrectly. For measuring RTs, we included only data from trials in which participants gave correct responses to the Go stimuli. Finally, for each participant, we obtained the averaged RTs and correct rates separately for the Go stimuli items in the Go/Nogo and Stroop groups.

Using SPSS 22.0, we performed 2 (task type: Go/Nogo, Stroop-like) × 3 (statement type: TTFF, TF, FT) repeated measures analysis of variance. The dependent variable was the response times (RTs) and accuracy of the participants to Go stimuli. We considered p < 0.05 to be statistically significant.

7 – The results are mixed with the statistics, and as no previous section was organized, it was difficult to follow the section. I strongly suggest an improved organization, with declared assumption checks, tests employed to assess the significant differences, effect sizes, etc.  

Response: We thank the reviewer for this suggestion. Combine Q6, we have made modifications to the corresponding sections to make them clearer. Please see Pages 7—9, line 284 -305, the Statistical analysis and Behavioral results section.

8 – The authors must be aware of all limitations of the study. I suggest you to insert a paragraph to describe and fully discuss them.

Response: Thank you for your valuable suggestions. Based on your suggestion, we have added the limitations section. Please see Pages 13—14, line 499-523,  Limitations section:

Limitations

This study faces several limitations that should be considered when interpreting its findings and designing future research. Firstly, the dependence on a sample of students may not generalize across different demographic groups. Additionally, the experimental setup, which focuses on speeded reasoning tasks, might not accurately replicate the natural pacing of real-world scientific reasoning, potentially biasing results towards those proficient under time constraints. Moreover, the study’s narrow focus on specific types of inhibitory tasks (Go/Nogo and Stroop-like) may not capture all relevant aspects of cognitive inhibition involved in scientific reasoning, suggesting the need for broader investigatory approaches in future research. Moreover, this study only focused on the semantic inhibitory effect of DLPFC in scientific reasoning. However, many studies have reported the role of VLPFC in semantic inhibition (e.g., [37]), but our research did not focus on the role of VLPFC in scientific reasoning. Further research can take into account the role of VLPFC.

Furthermore, the study's conditions of time pressure and sophisticated task/conceptual information highlight the roles of response inhibition and semantic inhibition in scientific reasoning. Teachers should consider presentation methods in science instruction to optimize inhibition processes. While multiple conceptual domains have been explored to understand the relationship between a single conceptual domain, different statements, and inhibition types, future research should focus on examining a single conceptual domain for more detailed insights. Additionally, the experimental materials used in this study had limited trials for certain concepts within a specific field. Therefore, it is recommended that future researchers increase the number of trials and reduce the number of conceptual fields to enhance the reliability and validity of the findings.

9 – I strongly advise you to insert the raw data into a public online repository.

Response: Thank you for your helpful suggestion. Once the manuscript was accepted, we will upload the data to a public online repository.

Finally, we appreciate your constructive comments. We have gained valuable insights from them, which have significantly enhanced the quality of our manuscript.

Best regards,

Author

Round 2

Reviewer 1 Report

Comments and Suggestions for Authors

I believe that the paper is ready to be published.

Comments on the Quality of English Language

Minor editing

Author Response

Dear Reviewer:

We are highly thankful to the worthy editor and honourable reviewers for providing us with an opportunity to publish the manuscript (brainsci-2994060, Neural Mechanisms of Inhibition in Scientific Reasoning: Insights from fNIRS). We appreciate your constructive comments during the manuscript revision process, and they significantly improved the quality of our manuscript.

Best regards,

Author

Reviewer 3 Report

Comments and Suggestions for Authors

Dear authors, thank you for your valuable attention to my comments. I have no further doubts.

My best wishes.

Author Response

We are highly thankful to the worthy editor and honourable reviewers for providing us with an opportunity to publish the manuscript (brainsci-2994060, Neural Mechanisms of Inhibition in Scientific Reasoning: Insights from fNIRS). We appreciate your constructive comments during the manuscript revision process, and they significantly improved the quality of our manuscript.

Best regards,

Author